# Evoking Generalized Cognition for Exemplar Free Continual Learning via Granular Ball Representation

## Abstract

Catastrophic forgetting remains a critical challenge in deep learning, particularly when samples from previously encountered classes are unavailable. This challenge drives advances in Exemplar-Free Class-Incremental Learning (EFCIL). However, incremental learning approaches typically use a single, fixed granularity for class representation, such as prototypes or features. We show that class representations exhibit varying granularity both within and across tasks, with the granularity of new classes gradually increasing as tasks progress, potentially leading the model to bias toward new classes during classification. To address this, we propose Granular Ball Incremental Learning (BallIL), which uses granular ball representation for multi-granularity class description and progressively expands the granularity of old classes to balance inter-task differences. Based on class concepts provided by the granular ball representation, we design concept-informed representation and decision uncertainties to assess loss for classification tasks. To address the issue of outdated class representations in new tasks due to feature drift, we develop a Synergy Drift Estimation (SDE) module for BallIL, ensuring that past concepts remain effective in the new representation space. Our extensive experiments across six datasets consistently highlight the superior performance of our method compared to current state-of-the-art methods. The code will be released.

## 1 Introduction

Deep learning has demonstrated exceptional success across diverse applications (He et al., 2016; Paissan et al., 2024), relying predominantly on the predefinition of all classes during training. However, when processing dynamic data from emerging classes, neural networks experience catastrophic forgetting (French, 1999), where previously learned knowledge is lost as new information is assimilated. This challenge has spurred research into the continuous learning capabilities of deep neural networks, a paradigm known as Continual Learning or Incremental Learning (Xu et al., 2025; Liu & Chang, 2025; Xuan et al., 2024).

In incremental learning, the complete dataset from previous classes is unavailable due to the streaming nature of the data, prompting the adoption of a compromise strategy, namely the rehearsal-based method (Cha et al., 2023; Liu et al., 2021; Castro et al., 2018; Rebuffi et al., 2017). This approach retains a small set of exemplars from old classes and replays them in subsequent sessions to counteract forgetting. However, exposing raw data raises privacy concerns, and the growing number of exemplars imposes greater storage demands on the system, necessitating the exploration of Exemplar-Free Class Incremental Learning (EFCIL) approaches (Cotogni et al., 2025; Wang et al., 2025; Goswami et al., 2024). In the EFCIL, previous data is tricky to harness for training a model capable of distinguishing between old and new classes, as the model doesn't even know which data corresponds to the old classes exactly. Despite challenges, EFCIL has gained momentum owing to its pragmatism, and this work focuses on its forgetting mechanism.

A common approach to handling EFCIL is to retain previous prototypes for synthesizing features used in pseudo-rehearsal (He et al., 2025; Nori et al., 2025; Masana et al., 2023), resembling rehearsal-based methods. However, these methods focus on data representation at a single, fixed

granularity, lacking consideration for the multi-granularity characteristics of classes, as illustrated in Fig. 1. Based on an intriguing observation in Fig. 2, the distribution regions of class data in the representation space vary in size, which occurs not only within an individual task (intra-task granularity differences) but also across multiple tasks (inter-task granularity differences). Specifically, the average granularity tends to increase as tasks progress. Unlike intra-task granularity differences, often caused by inconsistent variance in class data, inter-task variations may be inherently tied to the incremental learning setting. This potentially biases the model toward new classes with larger granularity during classification, thereby exacerbating forgetting. Therefore, a plausible conjecture is that adopting a multi-granularity perspective and progressively generalizing past representations in EFCIL may help improve model performance. This essentially evokes the model's generalized cognition of old classes.

Inspired by the emerging granular ball theory (Xie et al., 2025; 2024b; Liu et al., 2024), which has been widely explored for its adaptive multi-granularity representation of class data, we incorporate it into deep incremental learning (Zheng et al., 2024; Marczak et al., 2025) and develop *Granular Ball Incremental Learning* (BallIL) as a feasible solution for EFCIL. In BallIL, we use granular balls to capture intra-task granularity differences in class representation. As incremental tasks progress, a blur factor is introduced to gradually expand the representation range of granular balls for old classes, mitigating inter-task granularity differences. Focusing on specific classification tasks, we propose concept-informed representations and decision uncertainty based on class concepts defined by granular balls. This prevents new classes from eroding old class concepts in the embedding space and ensures that the classification layer can recognize old classes. Furthermore, to preserve the ef-

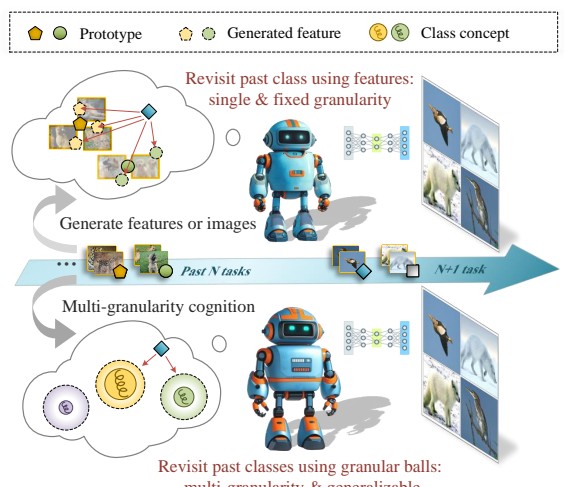

Figure 1: Top: Previous approaches revisit old classes using concrete features with a fixed and single granularity at the finest level. Bottom: Our approach enables multi-granularity representations of past classes and supports generalization.

fectiveness of past classes in the new model, we propose a Synergy Drift Estimation (SDE) module that dynamically updates granular ball representations during training.

The main contributions of this paper are as follows:

- To the best of our knowledge, this is the first work to address the EFCIL problem from a multi-granularity perspective, highlighting how increasing class granularity with task progression may worsen forgetting.
- Granular Ball Incremental Learning (BallIL) is developed using granular ball representations, with concept-informed representation and decision uncertainty evaluations designed for incremental classification.
- We design the Synergy Drift Estimation (SDE) module for BallIL, enabling the dynamic updating of past class concepts during training for effective representation in the new embedding space.

## 2 RELATED WORK

**Exemplar-Free Class Incremental Learning**. EFCIL has gained significant attention for its advantages in privacy protection and storage efficiency (Xu et al., 2025; Liu & Chang, 2025; Luo et al., 2024; Rypeść et al., 2024). A major challenge is the lack of knowledge about the true distribution of past classes, which diminishes the model's ability to discriminate between new and old data, exacerbating forgetting. To tackle this issue, SDC (Yu et al., 2020) uses interpolation to estimate the

drift of class prototypes in the new representation space and employs a Nearest Class Mean (NCM) classifier, which is minimally impacted by forgetting problem. Such a drift estimation approach has inspired a series of subsequent EFCIL methods (Zhu et al., 2025; Li et al., 2024; Toldo & Ozay, 2022). While NCM-based methods significantly boost EFCIL, they in fact circumvent the inherent limitations of the unified classifier. One approach to enhancing the effectiveness of the unified classifier is to augment the prototypes (Zhu et al., 2025; Malepathirana et al., 2023; Petit et al., 2023). However, these methods focus on the finest-grained sample features, neglecting the multi-granularity nature of class representations. At this juncture, this work argues that class granularity across different tasks should be balanced in order to combat forgetting.

**Granular Ball Theory**. Granular ball theory provides an important multi-granularity approach to data covering, originating from classification tasks (Xie et al., 2024a; Zhang et al., 2023) and gradually demonstrating its potential in clustering (Xia et al., 2022) and feature selection (Cao et al., 2024). Due to its excellent ability to adaptively characterize data distributions, it has recently been widely applied to multi-granularity data representation (Xie et al., 2024b; Huang et al., 2025), a class of methods referred to as Granular Ball Representation. It employs a self-supervised approach to construct a series of balls of varying sizes, enabling coarse-grained descriptions of sample distributions, with each ball enveloping multiple samples to represent them at a coarser granularity. By leveraging granular ball representation, reinforcement learning is advanced through cognitive generalization (Liu et al., 2024), with granular balls employed to characterize data distributions in a cognitive latent space. Distinct from previous methods, our work leverages the multi-granularity properties of granular balls to construct multi-granularity concepts for deep class incremental learning.

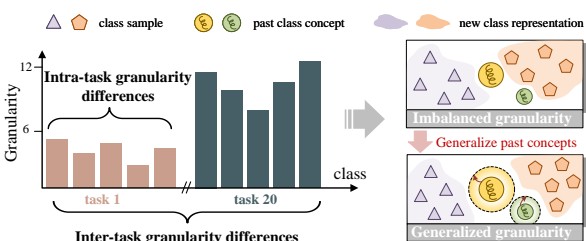

Figure 2: The bar chart depicts the knowledge granularity for each class across 20 tasks in the CIFAR-100 dataset. Each task includes data from five categories, with the chart focusing on the first and last tasks. Granularity is characterized by the Euclidean distance from the class center to the farthest sample of the class. The granularity difference among classes is relatively small within a single task, but it increases as tasks progress.

## 3 METHODOLOGY

### 3.1 PRELIMINARIES AND OVERVIEW

**Preliminaries**. The dynamic data stream is represented as $\mathcal{D} = \{D^t\}_{t=1}^T$, where $T$ denotes the total number of learning phases. For the training data $D^t$ in the $t$-th task, the candidate label set $C^t$ comprises entirely new classes, and $C^t \cap C^i = \emptyset \, (1 \le i < t)$. The dataset $D^t$ is defined as $D^t = \{X^t, Y^t\}$, where $X^t = \{x_i\}_{i=1}^{n_t}$ denotes the image set, $Y^t = \{y_i \in C^t\}_{i=1}^{n_t}$ denotes the target label set, and $n_t$ is the number of samples in $D^t$. During training phase $t$, the data is used to train a feature extractor $f^t : \mathbb{R}^{h \times w \times 3} \to \mathbb{R}^d$ and a unified classifier $g^t : \mathbb{R}^d \to \mathbb{R}^{p_t}$, where $d$ is the dimension of embedding features, and $p_t = \sum_{i=1}^t |C^i|$ denotes the total number of classes encountered up to phase $t$. Commonly, $g^t$ is a linear classifier with a weight matrix $\mathbf{W} \in \mathbb{R}^{p_t \times d}$, and the model output can be expressed as $h^t(x) = \sigma(\mathbf{W} f^t(x))$, where $\sigma$ is the softmax function.

**Overview**. The proposed method is illustrated in Fig. 3. We use a classification network with knowledge distillation as the baseline. At the end of each task, we generate and store multi-granularity class concepts using granular balls, which are progressively generalized as tasks advance. To maintain the effectiveness of old class concepts in the new embedding space, we design a Synergy Drift Estimation module to activate them. Based on the activated concepts, we introduce Concept-informed Representation and Decision Uncertainty loss ($\mathcal{L}_{ru}, \mathcal{L}_{du}$) to build the incremental model.

### 3.2 MULTI-GRANULARITY CONCEPT REPRESENTATION WITH GRANULAR BALL

As discussed earlier, class representations exhibit granularity differences, which motivates the introduction of granular balls to offer multi-granularity and generalizable representations. Suppose

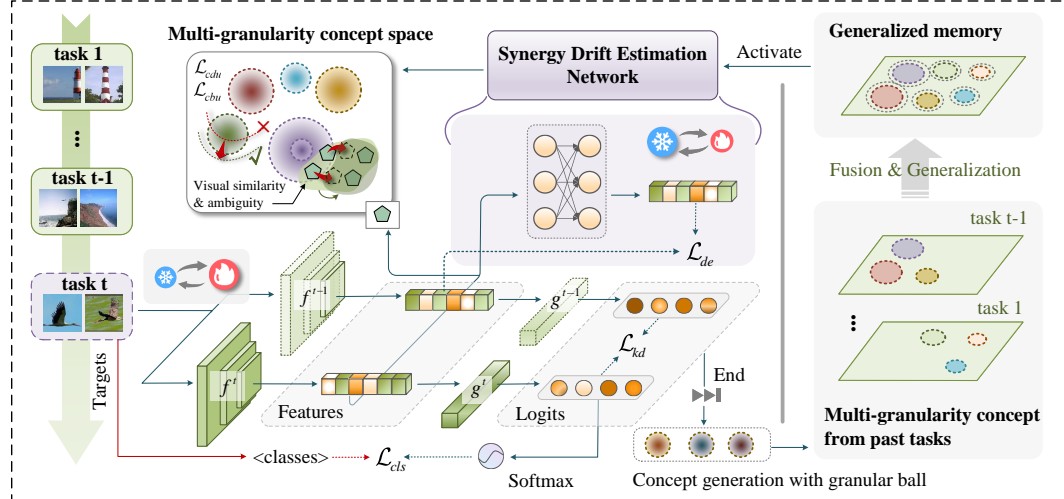

Figure 3: The overview of BallIL. Granular ball is abbreviated as GB. The method is implemented on a ResNet-18, and employs knowledge distillation as the baseline model.

$\mathbf{GB}^t = \{\boldsymbol{gb}_1^t, \boldsymbol{gb}_2^t, \ldots, \boldsymbol{gb}_m^t\}$ represents the set of granular balls generated at the end of task $t$, where $\forall \boldsymbol{gb}_i^t \in \mathbf{GB}^t$ indicates granular balls with varying granularity. Since the class distribution stabilizes at the end of a task and each granular ball represents the generalized concept of a single class, the samples ideally covered within the ball are expressed as $\mathbb{O}_{gb_i^t} = \{f^t(x_j) \mid \forall x_j \in X^t, y_j = c_i^t\}$. Wherein $c_i^t \in C^t$ corresponds to the class concept that $\boldsymbol{gb}_i^t$ characterized, the label of $\boldsymbol{gb}_i^t$ can be expressed as $\boldsymbol{l}_i^t = c_i^t$. Furthermore, the center $\boldsymbol{c}_i^t$ and the radius $\boldsymbol{r}_i^t$ of $\boldsymbol{gb}_i^t$ are defined as follows:

$$\boldsymbol{c}_i^t = \frac{1}{\mid \mathbb{O}_{gb_i^t} \mid} \sum_{z_j \in \mathbb{O}_{gb_i^t}} z_j, \ \boldsymbol{r}_i^t = \max_{z_j \in \mathbb{O}_{gb_i^t}} \{\| z_j - \boldsymbol{c}_i^t \|_2^2\}, \tag{1}$$

where $z_j = f^t(x_j)$ denotes the feature in the embedding space. Without the intricate process of granular ball splitting (Xie et al., 2024b), each ball is designated to represent a whole class concept, simplifying the computational process. The concept encapsulated by the ball is multi-granular and abstract, detached from specific samples, as each ball is defined by only three elements, i.e., $\boldsymbol{gb}_i^t = \{\boldsymbol{c}_i^t, \boldsymbol{r}_i^t, \boldsymbol{l}_i^t\}$. The generation of granular balls to achieve multi-granularity representation can be depicted by a standard model (Xie et al., 2024b), formulated as:

$$\min \frac{n_t}{\sum_{i=1}^m \mid \widetilde{\mathbb{O}}_{gb_i^t} \mid} + m, \ s.t. \ quality(gb_i^t) \geq T, \tag{2}$$

where $\widetilde{\mathbb{O}}_{gb_i^t} = \{\forall z_j \in f^t(X^t) \mid \| z_j - \boldsymbol{c}_i^t \|_2^2 \leq \boldsymbol{r}_i^t\}$ represents all the samples that could potentially fall within the coverage of the granular ball, and *quality* reflecting the proportion of samples that were labeled with class $\boldsymbol{l}_i^t$ in $\widetilde{\mathbb{O}}_{gb_i^t}$. As the number of samples $n$ and the number of concepts $m$ are fixed within a task in incremental learning, the above model can be regarded as a bi-level optimization problem related to $\mid \widetilde{\mathbb{O}}_{gb_i^t} \mid$ and $quality(gb_i^t)$. Since $\mid \widetilde{\mathbb{O}}_{gb_i^t} \mid \in [|\mathbb{O}_{gb_i^t}|, n_t]$ (proved in Appendix A) and $quality(gb_i^t) = |\mathbb{O}_{gb_i^t}| / \mid \widetilde{\mathbb{O}}_{gb_i^t} \mid$, the rapid maximization of $\mid \widetilde{\mathbb{O}}_{gb_i^t} \mid$ leads to a reduction in the objective function. However, it does not adequately address the inner optimization goal, which plays a more pivotal role in mitigating concept confusion. As a result, we place greater emphasis on the inner optimization $\min(T - quality(gb_i^t))$, where threshold $T$ is set to 1 to ensure no overlap between different granular balls. Inspired by contrastive learning (He et al., 2020), we achieve the above inner optimization by introducing the following loss in the incremental learning phase:

$$\mathcal{L}_{co} = -\frac{1}{n_t} \sum_{x_i \in X^t} \log \frac{\sum_{z_j \in S(z_i)} \exp(\mathcal{K}(z_i, z_j)/\mathcal{T})}{\sum_{z_k \in \overline{S}(z_i)} \exp(\mathcal{K}(z_i, z_k)/\mathcal{T})}, \tag{3}$$

in which, $\mathcal{T}$ represents the temperature, $\mathcal{K}$ denotes the Gaussian kernel, and $S(\cdot)$ and $\overline{S}(\cdot)$ indicate the sets of samples of the same and different classes, respectively.

To generate a multi-granularity concept space during the $t$-th learning phase, we propose integrating past $t-1$ concepts with a fusion strategy $\mathcal{GB}^t = \bigcup_{i=1}^{t-1} \Psi \left( \mathbf{GB}^i \right)$. Here, $\Psi$ denotes the operation of further generalizing past concepts. By introducing the blur factor $\psi$ to model the gradual blurred concepts, we define $\widetilde{\boldsymbol{gb}}_j^i = \Psi \left( \boldsymbol{gb}_j^i \right)$ for $\forall \boldsymbol{gb}_j^i \in \mathbf{GB}^i$ ($1 \leq i \leq t-1$) and calculate its radius as :

$$\widetilde{\boldsymbol{r}}_j^i = \boldsymbol{r}_j^i \cdot (1 + (t-i) \cdot \psi).\tag{4}$$

Following that, granular balls are activated using a synergy drift estimation module (discussed subsequently) to ensure the validity of generalized concepts in the new embedding space, with the activation operation for $\widetilde{\boldsymbol{gb}}_j^i$ formulated as $\mathcal{A}^{SDE} \left( \boldsymbol{c}_j^i \right)$. For simplicity, the activated granular balls will still be denoted by $\boldsymbol{gb}_j^i = \left\{ \boldsymbol{c}_j^i, \boldsymbol{r}_j^i, \boldsymbol{l}_j^i \right\}$ in the subsequent text.

### 3.3 Concept-Informed Uncertainty

**Concept-Informed Representation Uncertainty**. Intuitively, the phenomenon of forgetting in the representation space is irreversible, with the representations of past classes being overwritten by those of new classes. We attribute forgetting in the multi-granularity concept space to semantic ambiguity induced by visual similarity, where samples $z_j \in f^t \left( X^t \right)$ from class $c_i^t \in C^t$ are mistakenly placed within the representation scope of concept $\boldsymbol{gb} \in \mathcal{GB}^t$, significantly amplifying uncertainty in the representation space. Therefore, we optimize the feature extractor $f^t$ at session $t$ to construct an appropriate representation space without damaging the past multi-granularity concepts described by $\mathcal{GB}^t$. Consequently, we can construct an ambiguity sample set in the multi-granularity representation space that demonstrates semantic ambiguity for any given concept $\boldsymbol{gb}_i \in \mathcal{GB}^t$:

$$\mathbb{A}_{\boldsymbol{gb}_i} = \left\{ x_j \mid \forall x_j \in X^t, \| f^t \left( x_j \right) - \boldsymbol{c}_i \|_2^2 \leq \boldsymbol{r}_i \right\}.\tag{5}$$

Here, we omit the superscript of $\boldsymbol{gb}_i \in \mathcal{GB}^t$, which denotes the task indicator, as it can correspond to any of the past $t-1$ tasks. For an arbitrary granular ball, expected posterior probabilities can be defined with its ambiguity sample set:

$$\mathrm{P} \left( \boldsymbol{l}_i \mid \boldsymbol{gb}_i \right) = \underset{\boldsymbol{c}_i, \boldsymbol{r}_i, \mathbb{A}_{\boldsymbol{gb}_i}}{\mathbb{E}} \left[ \mathrm{P} \left( \boldsymbol{l}_i \mid \mathbb{A}_{\boldsymbol{gb}_i}, \boldsymbol{c}_i, \boldsymbol{r}_i \right) \right].\tag{6}$$

Ambiguous samples closer to a concept's central region generate greater semantic ambiguity, intensifying uncertainty in the representation space. To quantify the uncertainty of the concept, inspired by Shannon's work (Shannon, 1948), we define the self-information as:

$$\begin{aligned} \mathrm{I} \left( \boldsymbol{gb}_i \right) &= - \log \mathrm{P} \left( \boldsymbol{l}_i \mid \mathbb{A}_{\boldsymbol{gb}_i}, \boldsymbol{c}_i, \boldsymbol{r}_i \right) \\ &= - \log \frac{1}{\sum_{x_j \in \mathbb{A}_{\boldsymbol{gb}_i}} \exp \left( \frac{\boldsymbol{r}_i - \| \boldsymbol{c}_i - z_j \|_2^2}{\mathcal{T}} \right) + \epsilon}, \end{aligned}\tag{7}$$

where, $\epsilon$ is a small positive constant to prevent division by zero when $\mathbb{A}_{\boldsymbol{gb}_i} = \emptyset$. A higher self-information, as described in Eq. (7), indicates greater susceptibility of the concept to being forgotten. To account for diverse concepts, we employ entropy to quantify the uncertainty in the multi-granularity concept space and design a loss to mitigate representation uncertainty:

$$\mathcal{L}_{ru} = \frac{1}{\mid \mathcal{GB}^t \mid} \sum_{\boldsymbol{gb}_i \in \mathcal{GB}^t} \mathrm{P} \left( \boldsymbol{l}_i \mid \mathbb{A}_{\boldsymbol{gb}_i}, \boldsymbol{c}_i \right) \cdot \mathrm{I} \left( \boldsymbol{gb}_i \right).\tag{8}$$

By applying the above loss during the learning phase, the model is encouraged to distinguish between new and past classes, thereby preventing representation-level forgetting.

**Concept-Informed Decision Uncertainty**. Unlike the irreversible forgetting at the representation level, forgetting at the decision level can be moderated by refining the unified classifier head, provided the concepts remain separable within the representation space. The performance degradation due to decision-level forgetting primarily arises from task-wise class imbalance, which is especially pronounced in our exemplar-free setting, where no samples or features are replayed in the multi-granularity concept space. To tackle this issue, we propose modeling uncertainty at the decision level to quantify the degree of confusion among different classes. Denote the classifier head $g^t$ parameterized by $\mathbf{W} = \left\{ \mathbf{w}_1; \mathbf{w}_2; \ldots; \mathbf{w}_{p_t} \right\}$, where weight vector $\mathbf{w}_j \in \mathbb{R}^d$ is used to identify class

$j$ among all $p_t$ classes. Potentially, the set of classifiers susceptible to confusion with the concept $gb_i \in \mathcal{GB}^t$ during decision-making is expressed as:

$$\mathbb{C}_{gb_i} = \left\{ \mathbf{w}_j \mid \forall \mathbf{w}_j \in \mathbf{W}, \frac{\mid \mathbf{w}_j^T c_i + b_j \mid}{\sqrt{\sum_{s=1}^d w_s^2}} \leq r_i \right\}, \tag{9}$$

where $b_j$ represents the bias corresponding to the weight vector $\mathbf{w}_j$. Based on this, we define the probability of the $gb_i$ being correctly classified and introduce a decision-level self-information measure as follows:

$$\widetilde{\mathrm{I}}(gb_i) = -\log \mathrm{P}\left(l_i \mid \mathbb{C}_{gb_i}, c_i, r_i\right)$$
$$= -\log \frac{1}{\sup\limits_{\mathbf{w}_j \in \mathbb{C}_{gb_i}} \left\{ \exp\left( \left( r_i - \frac{\mid \mathbf{w}_j^T c_i + b_j \mid}{\sqrt{\sum_{s=1}^d w_s^2}} \right)/\mathcal{T} \right) + \epsilon \right\}}. \tag{10}$$

A higher self-information value reflects that the concept represented by $gb_i$ is more prone to confusion during decision-making. Accordingly, decision-level uncertainty over the entire multi-granular representation space is quantified using entropy, based on which a loss function is designed to reduce decision uncertainty:

$$\mathcal{L}_{du} = \frac{1}{\mid \mathcal{GB}^t \mid} \sum_{gb_i \in \mathcal{GB}^t} \mathrm{P}\left(l_i \mid \mathbb{C}_{gb_i}, c_i, r_i\right) \cdot \widetilde{\mathrm{I}}(gb_i). \tag{11}$$

### 3.4 SYNERGY DRIFT ESTIMATION MODULE

Another critical challenge in the exemplar-free scenario is the invalidation of previously preserved multi-granular concepts due to the continuous evolution of the feature extractor $f^t$. This underscores the pressing need to reactivate the past concepts to ensure their effective involvement in training for the novel session. Estimating the drift of old classes in the new representation space is a viable approach (Yu et al., 2020; Gomez-Villa et al., 2025), but its reliance on a fully trained model makes incorporating old class representations into new task learning impractical. This indicates a typical chicken-and-egg problem between incremental learning and drift estimation. Facing with such dilemma, we propose the Synergy Drift Estimation (SDE) module with a joint-cross training strategy. Concretely, we divide drift estimation and incremental training into two alternating phases:

- In the *incremental learning phase*, the SDE model is frozen, and the past concept $gb_i \in \mathcal{GB}^t$ is activated as:

$$r_i \leftarrow \mathcal{A}^{SDE}(r_i). \tag{12}$$

- In the *SDE learning phase*, the feature extractor $f^t$ is frozen, and the SDE model, consisting of a single linear layer, is trained with the mean squared error:

$$\mathcal{L}_{de} = \frac{1}{n_t} \sum_{x_i \in X^t} (f^t(x_i) - \mathcal{A}^{SDE}(f^{t-1}(x_i)))^2. \tag{13}$$

The above training phases are crossed within a single incremental task, allowing the multi-granularity concepts to be synergistically updated for the new task learning.

### 3.5 OVERALL OPTIMIZATION

Similar to most mainstream incremental learning methods (Shi & Ye, 2023; Li et al., 2024), LwF (Li & Hoiem, 2018) is used as the baseline method, integrating cross-entropy for learning new classes with knowledge distillation to preserve previously acquired knowledge. In the incremental learning phase, the overall loss is expressed as:

$$\mathcal{L}_{inc} = \mathcal{L}_{base} + \alpha \mathcal{L}_{co} + \gamma \mathcal{L}_{ru}, \tag{14}$$

where $\alpha$, and $\gamma$ are weighting parameters. After the training phase, the representation space is preserved while the classification layer is adjusted with the following loss function:

$$\mathcal{L}_{cla} = \mathcal{L}'_{base} + \delta \mathcal{L}_{du}, \tag{15}$$

where $\delta$ indicate the regularization strength and $\mathcal{L}'_{base}$ refers to the baseline without the inclusion of the knowledge distillation term, and treat ball centers of past concepts as classification data to construct the loss $\eta \mathcal{L}_{gb}$ for $g^t$ correction. The released code will include training details.

Table 1: Results (%) on CIFAR-100, TinyImageNet, and ImageNet-Subset. The highest recorded value is highlighted in bold, whereas the second-best result is underlined for emphasis. Methods identified by † are obtained from (Goswami et al., 2024).

| | Method | CIFAR-100 | | TinyImageNet | | ImageNet-Subset | |
|---|---|---|---|---|---|---|---|
| | | $A_{last}$ | $A_{inc}$ | $A_{last}$ | $A_{inc}$ | $A_{last}$ | $A_{inc}$ |
| 5 phases | LwF† | 45.35 | 61.94 | 38.81 | 49.7 | 50.88 | 69.11 |
| | PASS | 48.15 ± 0.88 | 63.32 ± 0.72 | 34.78 ± 0.39 | 45.55 ± 0.56 | 39.27 ± 0.05 | 59.87 ± 0.26 |
| | SSRE | 37.30 ± 1.31 | 51.86 ± 0.31 | 28.75 ± 0.10 | 40.76 ± 1.34 | 35.16 ± 0.38 | 50.24 ± 0.74 |
| | FeTrIL | 41.96 ± 0.34 | 59.21 ± 0.51 | 21.50 ± 0.77 | 35.25 ± 1.79 | 41.86 ± 1.27 | 57.93 ± 1.42 |
| | FeCAM† | 47.28 | 61.37 | 25.62 | 39.85 | 54.18 | 67.21 |
| | NAPA-VQ | 46.68 ± 0.34 | 61.50 ± 0.07 | 26.67 ± 0.15 | 40.50 ± 0.53 | 38.93 ± 0.67 | 55.22 ± 0.67 |
| | FKSR | 44.50 ± 0.69 | 60.96 ± 0.10 | 32.29 ± 3.98 | 44.50 ± 3.64 | 47.73 ± 0.87 | 62.61 ± 0.38 |
| | FCS | 45.47 ± 3.00 | 61.93 ± 1.83 | 40.85 ± 0.40 | 54.07 ± 0.80 | 52.29 ± 1.22 | 66.94 ± 1.52 |
| | BallIL | **55.89 ± 0.30** | **68.49 ± 0.43** | 41.04 ± 0.12 | 54.48 ± 1.54 | **58.89 ± 1.17** | **70.61 ± 2.46** |
| 10 phases | LwF† | 26.14 | 46.14 | 27.42 | 38.77 | 37.9 | 61.6 |
| | PASS | 25.33 ± 4.51 | 43.14 ± 4.68 | 25.07 ± 0.28 | 36.01 ± 0.16 | 26.70 ± 0.22 | 49.03 ± 0.34 |
| | SSRE | 18.64 ± 1.43 | 33.15 ± 1.46 | 20.78 ± 1.01 | 33.73 ± 2.08 | 23.60 ± 0.28 | 39.30 ± 0.79 |
| | FeTrIL | 32.38 ± 0.70 | 50.26 ± 0.15 | 15.42 ± 0.64 | 27.91 ± 2.30 | 30.45 ± 1.14 | 48.06 ± 1.18 |
| | FeCAM† | 33.82 | 48.58 | 23.21 | 35.32 | 42.68 | 57.45 |
| | NAPA-VQ | 37.07 ± 0.12 | 51.94 ± 0.31 | 20.37 ± 0.06 | 31.72 ± 0.04 | 26.15 ± 0.75 | 42.35 ± 0.84 |
| | FKSR | 31.25 ± 2.01 | 48.43 ± 2.08 | 19.03 ± 0.24 | 32.99 ± 2.67 | 35.65 ± 0.38 | 52.47 ± 0.06 |
| | FCS | 32.95 ± 5.08 | 47.60 ± 6.23 | 30.71 ± 2.85 | 45.19 ± 2.67 | 40.39 ± 1.70 | 56.12 ± 1.67 |
| | BallIL | **45.14 ± 0.71** | **59.40 ± 0.05** | **32.04 ± 1.67** | **46.49 ± 1.99** | **48.67 ± 0.23** | **64.94 ± 0.29** |

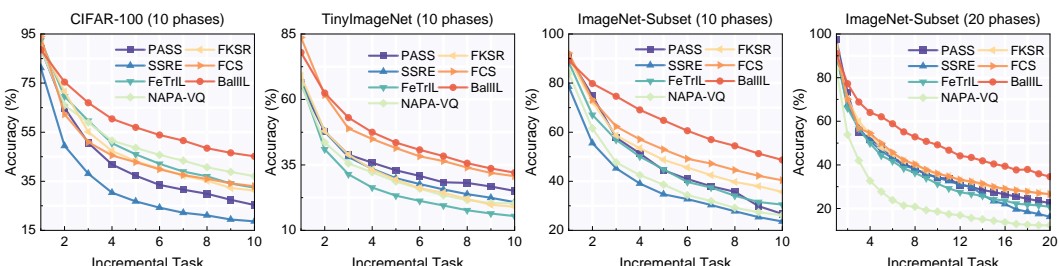

Figure 4: Classification accuracy at each incremental step on three datasets.

## 3.6 EXPERIMENTAL SETTING

**Datasets**. To demonstrate the performance of the proposed method, experiments are conducted on six widely used datasets: CIFAR-100 (Krizhevsky, 2009), TinyImageNet (Le & Yang, 2015), ImageNetSubset (Deng et al., 2009b), CUB-200 (Wah et al., 2011), Stanford Cars (Krause et al., 2013) and ImageNet-1K (Deng et al., 2009a). In the experiments, all classes within the datasets are equally divided into several phases, which indicates a more challenging cold-start setting (Goswami et al., 2024) that tests the model's ability to adapt and update data representations as tasks evolve. To reduce the effect of class order, three random seeds are used and the mean results are reported.

**Implementation Details**. As a widely adopted backbone network, our approach is implemented based on ResNet-18 (He et al., 2016) and trained from scratch, following (Zhu et al., 2021). The experiments are conducted on the PyCIL framework (Zhou et al., 2023). More implementation details are provided in the Appendix B.3. The code of BallIL will be made publicly available.

**Evaluation Metrics**. Following prior research (Zhai et al., 2024), we use two commonly adopted metrics to evaluate the experimental performance, including last task accuracy $A_{last}$ and average incremental accuracy $A_{inc}$. The former computes the average classification accuracy over all encountered samples upon completion of the last task. The average incremental accuracy represents the average output accuracy across all preceding tasks, and this metric better reflects the model's overall performance across all tasks.

Table 2: Ablation results (%) for the components of the BallIL. The evaluation on three datasets with a 10-phase partition reports both the last task accuracy and the average incremental accuracy.

| | PC | CO | RU | DU | CIFAR-100 | | TinyImageNet | | ImageNet-Subset | |
|---|---|---|---|---|---|---|---|---|---|---|
| | | | | | $A_{last}$ | $A_{inc}$ | $A_{last}$ | $A_{inc}$ | $A_{last}$ | $A_{inc}$ |
| (a) | ✓ | | | | 11.73 | 28.98 | 14.58 | 23.14 | 28.74 | 36.17 |
| | ✓ | | | | 40.32 | 52.15 | 24.76 | 35.91 | 39.58 | 50.43 |
| (b) | ✓ | | ✓ | | 43.82 | 54.81 | 28.77 | 39.51 | 41.47 | 53.74 |
| | ✓ | | | ✓ | 43.66 | 55.14 | 29.86 | 42.25 | 44.53 | 52.94 |
| (c) | ✓ | ✓ | ✓ | | 44.71 | 56.75 | 30.84 | 45.72 | 47.43 | 62.17 |
| | ✓ | ✓ | | ✓ | 44.80 | 58.67 | 30.13 | 43.57 | 46.73 | 58.95 |
| (d) | ✓ | ✓ | ✓ | ✓ | **45.14** | **59.40** | **32.04** | **46.49** | **48.67** | **64.94** |

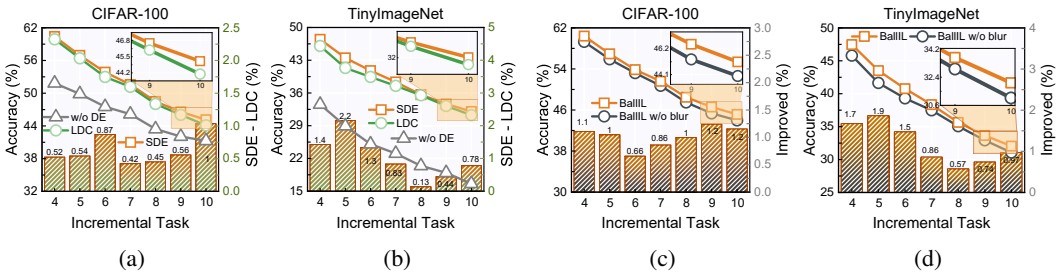

(a)        (b)        (c)        (d)

Figure 5: (a)-(b): ablation analysis of SDE on each incremental task. (c)-(d): comparison of BallIL and its variant without the blur factor ($\psi = 0$) across tasks. Left Y-axis (line chart) shows task accuracy, and right Y-axis (bar chart) shows BallIL's improvement over the second-best method.

## 3.7 COMPARISON WITH SOTA

We compare our BallIL method with the state-of-the-art EFCIL algorithms, including LDC (Gomez-Villa et al., 2025), FCS (Li et al., 2024), FKSR (Zhai et al., 2024), NAPA-VQ (Malepathirana et al., 2023), FeCAM (Goswami et al., 2023), FeTrIL (Petit et al., 2023), SSRE (Zhu et al., 2022), PASS (Zhu et al., 2021), and LwF (Li & Hoiem, 2018). To ensure consistent experimental conditions, we re-implemented the comparative methods based on an equal division of the dataset. The experiments are conducted with three random seeds, and both the mean and standard deviation are reported.

The experimental results in Table 4 demonstrate that BallIL consistently outperforms existing methods across CIFAR-100, TinyImageNet, and ImageNet-Subset in both 5-phase and 10-phase settings. BallIL achieves significant improvements over the second-best methods, with notable gains in last task accuracy and average incremental accuracy. In the 5-phase setting, BallIL shows improvements of approximately 16.1 and 8.2 in last task accuracy and average incremental accuracy on CIFAR-100, and 8.7 and 2.2 on ImageNet-Subset, respectively. Methods like FKSR and FCS show limited effectiveness, likely because they rely on initial tasks with more classes to learn generalized features. With equal class distribution, they cannot adequately update old class representations, leading to progressive obsolescence and error accumulation. Additionally, Fig. 4 shows BallIL consistently outperforming all baselines as tasks progress. The 20-phase results are given in the Appendix C.

## 3.8 ABLATION STUDY

**Results Analysis**. The effectiveness of the proposed BallIL method is validated through ablation studies on its components, with the results on three presented in Table 2. The proposed BallIL consists of four modules, including past concepts classification (PC), contrastive loss (CO), concept-informed representation uncertainty (RU), and concept-informed decision uncertainty (DU). Next, we analyze the CIFAR-100 results. According to ablation-a in Table 2, the incorporation of PC significantly improves model accuracy, indicating that using granular ball centers as representative features for classifier correction effectively mitigates bias. In ablation-b, introducing RU and DU improves $A_{inc}$ by 2.66 and 2.99, respectively. Furthermore, the integration of the CO yields more

Table 3: Results (%) on fine-grained datasets (CUB-200, Stanford Cars) and the large-scale dataset ImageNet-1K. All classes are evenly distributed across training phases.

| Method | CUB-200 | | | | Stanford Cars | | | | ImageNet-1K | |
|---|---|---|---|---|---|---|---|---|---|---|
| | 5 phases | | 10 phases | | 7 phases | | 14 phases | | 20 phases | |
| | $A_{last}$ | $A_{inc}$ | $A_{last}$ | $A_{inc}$ | $A_{last}$ | $A_{inc}$ | $A_{last}$ | $A_{inc}$ | $A_{last}$ | $A_{inc}$ |
| PASS | 34.04 | 49.00 | 26.37 | 41.08 | 20.71 | 37.13 | 12.30 | 25.46 | 33.95 | 48.21 |
| FeTrIL | 54.66 | 67.45 | 49.09 | 62.42 | 36.92 | 54.09 | 34.29 | 50.41 | 26.64 | 40.02 |
| FeCAM | 53.47 | 66.39 | 51.78 | 64.97 | 40.64 | 56.24 | 37.50 | 52.78 | 27.24 | 40.67 |
| ADC | 64.46 | 73.49 | 57.97 | 68.91 | 54.86 | 67.07 | 45.07 | 61.39 | - | - |
| BallIL | **65.31** | **74.06** | **58.25** | **69.68** | **55.13** | **67.87** | **45.25** | **63.99** | **36.84** | **50.37** |

improvements, enhancing the last task accuracy by 0.89 and 1.14 for RU and DU, respectively, which can be shown in ablation-c. Finally, ablation-d presents BallIL's full performance, showing that combining both modules achieves greater gains than using either alone. Overall, compared to the model considering only PC in Ablation-a, BallIL improves $A_{last}$ and $A_{inc}$ by 4.82 and 7.25, respectively, indicating that the synergy of all modules contributes to overall model enhancement.

**Effectiveness of SDE**. Our proposed BallIL method employs SDE module to continuously update the representation of old classes in the new embedding space. This section validates the effectiveness of SDE by comparing it with the LDC (Gomez-Villa et al., 2025), which shares the same network as SDE but updates old class knowledge only at the end of each task. Figs. 5(a)-(b) demonstrate that SDE's continuous updating of class representations during training cycles substantially outperforms LDC. The marked improvement over the baseline without drift compensation (DE) suggests that the feature extractor's adaptation to new tasks progressively diminishes the relevance of past knowledge.

**Effectiveness of Blur Factor**. In Eq. (4), we introduce the blur factor into the BallIL model, aiming to mitigate the issue of class granularity imbalance by expanding the old class representation characterized by granular balls. To verify the effect of the blur factor, we ablate it and present the performance differences across tasks in Figs. 5(c)-(d). The experimental findings demonstrate that addressing the inter-task knowledge granularity imbalance contributes to enhanced incremental learning efficacy in most cases. The impact of the blur factor on model performance is discussed in the Appendix D.2.

### 3.9 EXPERIMENTS ON FINE-GRAINED AND LARGE-SCALE DATASETS

We further evaluate BallIL on the fine-grained datasets CUB-200 and Stanford Cars, as well as the large-scale dataset ImageNet-1K, to assess its performance across diverse scenarios. Following widely adopted protocols (Goswami et al., 2024), the model is first warmed up on ImageNet-1K to acquire basic feature extraction ability before learning the fine-grained tasks. Part of the fine-grained results is taken from (Goswami et al., 2024), and comparative results on ImageNet-1K are obtained from (Magistri et al., 2025). The results in Table 3 show that BallIL achieves competitive performance on both fine-grained and large-scale datasets, benefiting from its granularity-based ball representation that alleviates class imbalance across tasks.

## 4 CONCLUSION

In this paper, we propose BallIL, a novel method for EFCIL. The model is designed based on intra- and inter-task concept granularity differences and implements multi-granularity representation of class using granular ball theory. Without generating additional samples or features, it progressively expands the granularity of old class concepts to mitigate classification bias caused by concept granularity differences. Additionally, it introduces concept-informed representation and decision uncertainty to alleviate catastrophic forgetting. Meanwhile, a novel drift estimation method, SDE, is incorporated to enable real-time updates of past concepts during new phase learning. The superiority of BallIL is demonstrated over existing exemplar-free methods across six benchmark datasets.

## REPRODUCIBILITY STATEMENT

Efforts have been made to ensure the reproducibility of the experiments and results presented in this work. The implementation details of the proposed BallIL method, including network architectures, training protocols, and hyperparameter settings, are provided in the main text and further elaborated in the Appendix. The datasets used are publicly available, and the data preprocessing steps are described in the Appendix. The effects of key parameters, such as the blur factor and multi-granularity representation settings, are also analyzed in the Appendix. Additionally, the implementation of the method will be released, allowing for verification of the results reported in the paper. All referenced materials collectively support the reproducibility of the reported findings.

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

APPENDIX

# A    PROOF FOR MAIN PAPER

Let $\mathbf{GB}^t = \{\boldsymbol{gb}_1^t, \boldsymbol{gb}_2^t, \dots, \boldsymbol{gb}_m^t\}$ be a set of granular balls generated on task $t$. For $\forall \boldsymbol{gb}_i^t \in \mathbf{GB}^t$, the samples ideally covered within the ball are all labeled with the same classes, denoted as $\mathbb{O}_{gb_i^t} = \{f^t(x_j) \mid \forall x_j \in X^t, y_j = c_i^t\}$, where $c_i^t$ is the $i$-th class in the task $t$. Given the sample set $\widetilde{\mathbb{O}}_{gb_i^t} = \{\forall z_j \in f^t(X^t) \mid \| z_j - \boldsymbol{c}_i^t \|_2^2 \leq \boldsymbol{r}_i^t\}$ that actually covered by the granular ball, there exist $| \widetilde{\mathbb{O}}_{gb_i^t} | \in [|\mathbb{O}_{gb_i^t}|, n_t]$, where $n_t$ is the number of sample for task $t$.

**Proof:** Since ball radius is calculated as

$$\boldsymbol{r}_i^t = \max\{\| z_j - \boldsymbol{c}_i^t \|_2^2 \mid z_j \in \mathbb{O}_{gb_i^t}\} \tag{16}$$

which implies that the the ball radius $\boldsymbol{r}_i^t$ equals the distance from its center $\boldsymbol{c}_i^t$ to the farthest sample of the same class. Thus, we have

$$\begin{aligned} \mathbb{O}_{gb_i^t} &= \{f^t(x_j) \mid \forall x_j \in X^t, y_j = c_i^t\} \\ \Leftrightarrow \quad \mathbb{S}_{gb_i^t} &= \{\forall z_j \in f^t(X^t) \mid \| z_j - \boldsymbol{c}_i^t \|_2^2 \leq \boldsymbol{r}_i^t, y_j = c_i^t\} \end{aligned} \tag{17}$$

where $\mathbb{S}_{gb_i^t}$ represents the sample set labeled as class $c_i^t$ within the radius of the granular ball. For $\widetilde{\mathbb{O}}_{gb_i^t}$ it can be written as

$$\begin{aligned} \widetilde{\mathbb{O}}_{gb_i^t} &= \{\forall z_j \in f^t(X^t) \mid \| z_j - \boldsymbol{c}_i^t \|_2^2 \leq \boldsymbol{r}_i^t\} \\ &= \mathbb{S}_{gb_i^t} \cup \{\forall z_j \in f^t(X^t) \mid \| z_j - \boldsymbol{c}_i^t \|_2^2 \leq \boldsymbol{r}_i^t, y_j = \overline{c_i^t}\} \end{aligned} \tag{18}$$

where $\overline{c_i^t}$ represents the complement of class $c_i^t$. Since

$$0 \leq | \{\forall z_j \in f^t(X^t) \mid \| z_j - \boldsymbol{c}_i^t \|_2^2 \leq \boldsymbol{r}_i^t, y_j = \overline{c_i^t}\} | \leq | X^t | \tag{19}$$

it follows from Eqs. (17)-(18) that the following holds

$$| \mathbb{O}_{gb_i^t} | = | \mathbb{S}_{gb_i^t} | \leq | \widetilde{\mathbb{O}}_{gb_i^t} | \leq | X^t | \tag{20}$$

Therefore, we can infer that $| \widetilde{\mathbb{O}}_{gb_i^t} | \in [|\mathbb{O}_{gb_i^t}|, n_t]$, in which $n_t = | X^t |$. Proof completed.

# B    EXPLANATION OF DETAILS

## B.1    FORMULATION OF EVALUATION METRICS

Following most previous research (Goswami et al., 2024; Gomez-Villa et al., 2025), this work evaluates the performance of comparison algorithms with two metrics, i.e., last task accuracy $A_{last}$ and average incremental accuracy $A_{inc}$. The $A_{last}$ refers to the prediction accuracy on all encountered classes at the end of the last task, while $A_{inc}$ indicates the average of the accuracy after all tasks.

For a sequence of $T$ tasks, the last task accuracy at the end of the task $T$ is calculated as

$$A_{last}^T = \frac{1}{|\mathcal{D}_{test}^{1:T}|} \sum_{(x,y) \in \mathcal{D}_{test}^{1:T}} \mathbb{I}\left(\mathcal{F}^T(x) = y\right), \tag{21}$$

where $\mathcal{D}_{test}^{1:T}$ denotes the test data from the initial stage to stage $T$, $\mathcal{F}^T(x)$ denotes the output label assigned by the model at stage $T$ for sample $x$, and $\mathbb{I}$ is an indicator function that returns 1 if the condition is met and 0 otherwise. By default, the last task accuracy refers to the accuracy at the last training stage $T$, and thus, $A_{last}^T$ is denoted as $A_{last}$. Then, the average incremental accuracy $A_{inc}$ is calculated as

$$A_{inc} = \frac{1}{T} \sum_{t=1}^{T} A_{last}^t. \tag{22}$$

Table 4: Results (%) on CIFAR-100, TinyImageNet, and ImageNet-Subset. The highest recorded value is highlighted in bold, whereas the second-best result is underlined for emphasis. For methods identified by †, the corresponding values are obtained from (Magistri et al., 2024).

| | Methods | CIFAR-100 | | TinyImageNet | | ImageNet-Subset | |
|---|---|---|---|---|---|---|---|
| | | $A_{last}$ | $A_{inc}$ | $A_{last}$ | $A_{inc}$ | $A_{last}$ | $A_{inc}$ |
| 20 phases | LwF† | 17.44 ± 0.73 | 38.39 ± 1.05 | 15.02 ± 0.67 | 32.94 ± 0.54 | 18.64 ± 1.67 | 40.23 ± 0.43 |
| | PASS | 25.31 ± 0.18 | 42.55 ± 0.82 | 18.73 ± 1.43 | 32.01 ± 1.68 | 22.45 ± 1.11 | 39.55 ± 0.60 |
| | SSRE | 9.97 ± 0.35 | 22.23 ± 0.15 | 10.69 ± 1.20 | 21.80 ± 2.37 | 16.25 ± 1.05 | 31.15 ± 1.53 |
| | FeTrIL | 25.17 ± 2.57 | 42.69 ± 2.49 | 11.76 ± 0.22 | 22.66 ± 1.31 | 20.74 ± 1.24 | 36.63 ± 1.43 |
| | NAPA-VQ | 23.43 ± 0.39 | 37.77 ± 0.01 | 13.53 ± 0.02 | 22.77 ± 0.11 | 12.28 ± 0.14 | 24.38 ± 0.03 |
| | FKSR | 18.66 ± 0.70 | 35.26 ± 0.52 | 14.93 ± 2.63 | 25.65 ± 3.35 | 26.60 ± 0.98 | 41.27 ± 0.71 |
| | FCS | 13.16 ± 7.72 | 29.84 ± 7.61 | 13.77 ± 0.76 | 23.41 ± 0.55 | 26.61 ± 0.49 | 41.28 ± 0.34 |
| | BallIL | **36.55 ± 0.52** | **51.41 ± 0.64** | **21.27 ± 0.46** | **34.33 ± 1.63** | **34.62 ± 0.26** | **51.60 ± 1.44** |

## B.2 ILLUSTRATION OF DATASETS

The CIFAR-100 dataset (Krizhevsky, 2009) consists of 60,000 images, each with a size of 32x32 pixels, organized into 100 classes. Each class contains 500 training images and 100 test images. The Tiny-ImageNet dataset (Le & Yang, 2015) includes 200 classes, where each class is comprised of 500 training images, 50 validation images, and 50 test images, all with a resolution of 64x64 pixels. This dataset offers additional stages and incremental classes, making it useful for evaluating the sensitivity of different methods. The ImageNet-Subset is derived from ImageNet-1k (Deng et al., 2009b) by randomly selecting 100 classes using random seed 1993. Each class in this subset contains approximately 1,300 training images and 50 test images. The image size in ImageNet-Subset is 256x256, which is considerably larger compared to the other two datasets.

## B.3 IMPLEMENTATION DETAILS

For CIFAR-100, we adopt the augmentation policies of CIFAR-10 provided by PyCIL (Zhou et al., 2023) and apply them consistently across all approaches to maintain fairness in comparison. For all datasets, we adhere to the standard approach by applying random crop and horizontal flip to the training set, similar to (Gomez-Villa et al., 2025). For the initial task across all datasets, we follow PyCIL's setup, setting the learning rate to 0.1, momentum to 0.9, and weight decay to 0.0005, training for 200 epochs with a batch size of 128. The learning rate is decreased by a factor of 10 at epochs 60, 120, and 180. For subsequent tasks in CIFAR-100 and ImageNet100, the learning rate is adjusted to 0.05, whereas for TinyImageNet, it is set to 0.001. These tasks are trained for 100 epochs, with the learning rate dropping by 10 at epochs 45 and 90. The temperature scaling factor is fixed at 2, while the regularization strength knowledge distillation is configured as 10 for CIFAR-100 and TinyImageNet, and 5 for ImageNet100.

All datasets are trained with 200 epochs for the initial task and 100 epochs for subsequent tasks, using a fixed batch size of 128. During training, the model was optimized using SGD with a momentum of 0.9, where the weight decay is set to 5e-4 for the initial task and reduced to 2e-4 for subsequent tasks. The temperature $\mathcal{T}$ is set to 2, while the distillation strength $\beta$ is maintained at 10, following (Masana et al., 2023). Within the BallIL framework, the SDE module adopts a single-layer linear architecture with bias terms, optimized via Adam at a 1e-3 learning rate. The training procedure is interleaved with the backbone network, with the SDE executing 20-epoch training cycles whenever the backbone network completes 40, 60, and 80 epochs in each task.

## C ADDITIONAL RESULTS

**Results on 20 phases**. We further evaluate BallIL under a 20-task split and report its performance against other methods in Table 4. The results highlight the notable advantage of our approach over state-of-the-art baselines. Fig. 6 illustrates the trend of last-task accuracy as the number of tasks increases. The results indicate that the proposed method consistently achieves superior performance

Table 5: Evaluation of SDE via Comparison with LDC (Gomez-Villa et al., 2025) and ablation of drift estimation. Datasets are partitioned into ten tasks.

| SDE (Ours) | LDC | CIFAR-100 | | TinyImageNet | | ImageNet-Subset | |
|---|---|---|---|---|---|---|---|
| | | $A_{last}$ | $A_{inc}$ | $A_{last}$ | $A_{inc}$ | $A_{last}$ | $A_{inc}$ |
| - | - | 41.27 | 54.27 | 16.59 | 33.97 | 37.81 | 56.16 |
| | ✓ | 44.10 | 58.89 | 31.26 | 45.18 | 46.21 | 61.49 |
| ✓ | | **45.14** | **59.40** | **32.04** | **46.49** | **48.67** | **64.94** |

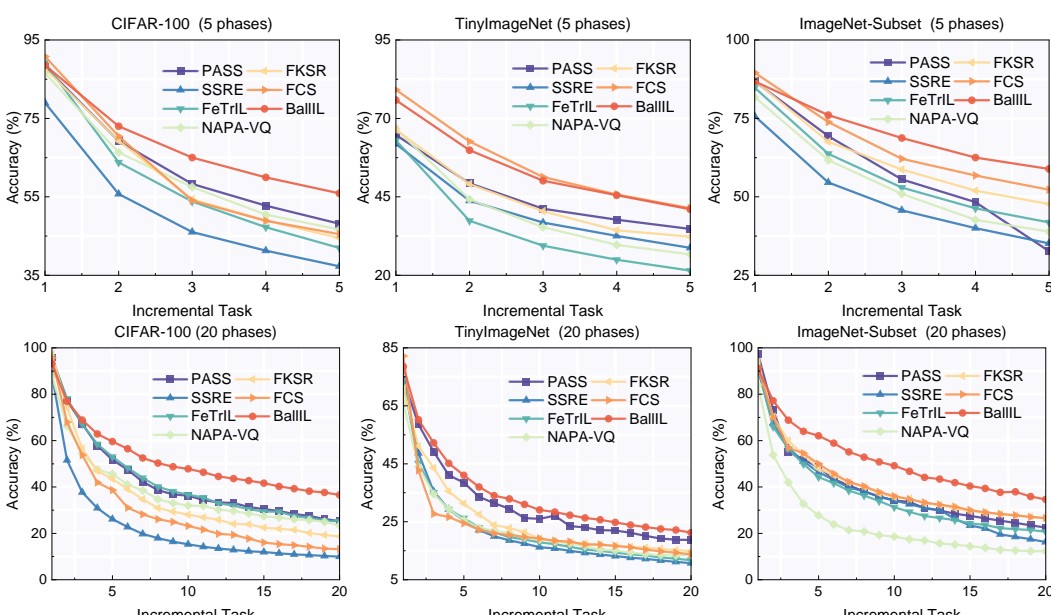

Figure 6: Classification accuracy after each incremental step on CIFAR-100, TinyImageNet, and ImageNet-Subset for 5 and 20 phases.

across all training stages in the exemplar-free setting. Additionally, Fig. 6 provides the task accuracy curves under the 5-phase division, serving as a supplement to the main paper.

**Effectiveness of SDE**. The proposed BallIL method incorporates a Synergy Drift Estimation (SDE) module to continuously refresh the representations of previously learned classes within the current embedding space. As shown in Table 5, the ongoing update of class representations by SDE throughout training significantly surpasses the performance of LDC. The notable gain relative to the baseline lacking drift compensation (DE) indicates that the feature extractor's adaptation to new tasks gradually reduces the relevance of prior knowledge.

## D ANALYSIS

### D.1 CHARACTERISTIC OF MULTI-GRANULARITY

This work discusses the phenomenon of intra- and inter-task granularity differences among class knowledge, which is demonstrated through empirical experiments in this section, as shown in Fig. 7. We partition the CIFAR-100 dataset into 10 and 20 tasks, respectively, and record the class knowledge granularity characterized by the granular ball radius at the end of each task. As shown in Fig. 7, the granularity of class knowledge varies significantly within a single task. This intra-task disparity may stem from the distribution characteristics of the data and the imbalance in the number of samples across different classes. Moreover, it is noteworthy that the average granularity of class knowledge tends to increase across tasks, as evidenced by the comparison between the first and last tasks. This phenomenon may be relevant to the gradual performance degradation observed

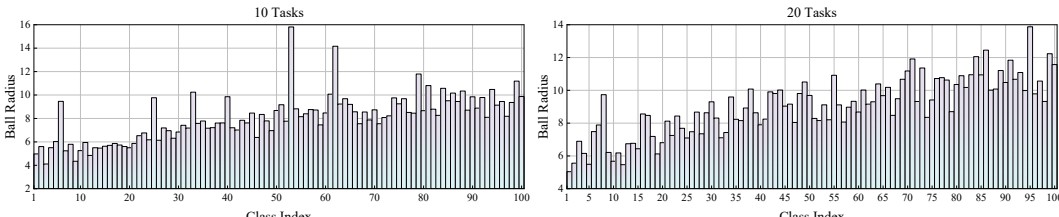

Figure 7: The class knowledge granularity at different tasks is characterized by the granular ball radius. The presented data is derived from experiments on CIFAR-100. The left figure illustrates the dataset partitioned into 10 tasks, whereas the right figure depicts the case with 20 tasks.

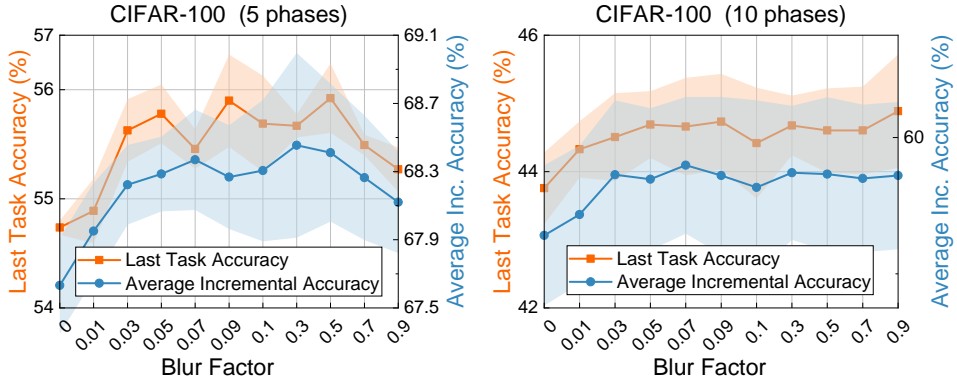

Figure 8: The effect of blur factor variations on classification accuracy in the CIFAR-100 dataset. The left figure shows the results for the dataset split into 5 tasks, while the right figure presents the case with 10 tasks. The shaded area in the figure represents the standard deviation. The left Y-axis represents last task accuracy, and the right Y-axis represents average incremental accuracy.

in new tasks during the incremental learning process. As the model must preserve knowledge of old classes while incorporating new ones, it hampers the model's ability to learn more compact and discriminative feature representations for the new classes. By comparing the experimental results of 10 and 20 tasks, it is evident that the increase in class knowledge granularity becomes more pronounced with more incremental tasks. This further highlights that, under the incremental learning setup, inter-task granularity differences of class knowledge are more likely to emerge.

## D.2 PARAMETER ANALYSIS

**Effectiveness of Blur Factor.** Based on our findings, a granularity discrepancy exists between tasks, where class knowledge in the latest incremental tasks typically exhibits larger granularity than that of older classes (refer to Fig. 7.). This discrepancy may bias the classifier toward new classes with larger granularity, thereby exacerbating forgetting. Therefore, this paper introduces the concept of granular balls to represent knowledge at multiple granularities and incorporates a blur factor to gradually expand the representation of old class knowledge. We investigate the impact of the blur factor on incremental learning performance using the CIFAR-100 dataset, with the experimental results presented in Fig. 8. Based on the results of the 5 tasks partition, as the value of the blur factor increases, the overall performance of the model shows an upward trend. This indicates that expanding the representation of old classes is effective in enhancing incremental learning. However, when the value of the blur factor exceeds 0.5, both the last task accuracy and the average incremental accuracy exhibit a noticeable decline. This indicates that this value exceeds the effective range and may lead to an overcorrection of the granularity of old class knowledge. In the experiment with 10 tasks, within the search range, an increase in the blur factor generally leads to an improvement in last task accuracy. This may be due to the larger granularity difference in class knowledge between tasks caused by more task divisions, which necessitates a larger blur factor to balance the granularity of class knowledge across different tasks.

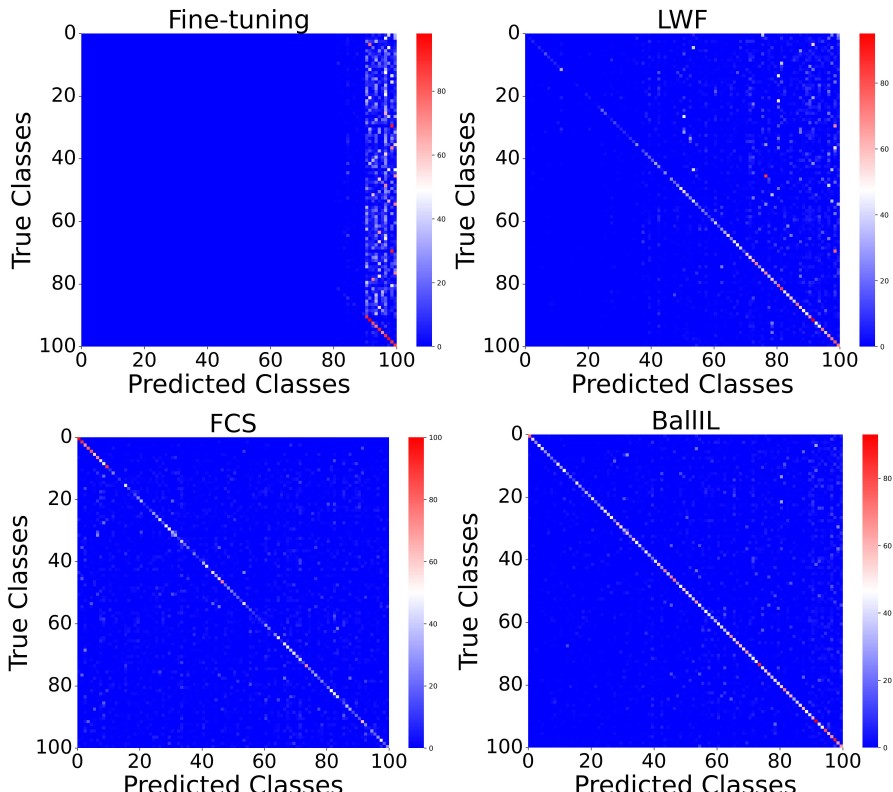

Figure 9: Confusion matrices for Fine-tuning, LWF, FCS, and BallIL on CIFAR-100. Both FCS and BallIL mitigate the task recency bias present in fine-tuning. Compared to FCS, BallIL exhibits more prominent red regions along the diagonal, indicating its superior ability to preserve past knowledge.

**Effectiveness of Trade-off Parameters**. The proposed BallIL consists of two stages during training. For the incremental learning stage, Fig. 10 shows that setting $\alpha$ too high or low hinders optimal performance, with the best value between 0.9 and 1. Based on this, the optimal $\gamma$ is 0.07. At the end of each task, the classification head is adjusted using Eq. (15), where $\eta$ controls the loss intensity for the classification of old concepts, while $\delta$ controls the decision uncertainty. As depicted in Fig. 10, a larger $\eta$ significantly improves prediction accuracy. In contrast, the impact of $\delta$ is less pronounced, with its optimal value lying between 0.3 and 0.7.

### D.3 COMPARISON OF CONFUSION MATRICES

Fig. 9 presents a comparison of confusion matrices derived from four different approaches: (1) conventional fine-tuning, where a model is incrementally trained using cross-entropy loss without applying any mechanisms to mitigate forgetting, (2) LwF, and (3) FCS. In these matrices, diagonal elements indicate correctly classified instances, whereas off-diagonal elements signify errors. The fine-tuning approach exhibits a strong bias towards recently learned tasks, as earlier classes are largely forgotten. In contrast, FCS and BallIL significantly mitigate this bias, enabling accurate classification of both previously seen and newly introduced categories. While these two approaches yield comparable outcomes, BallIL exhibits a more prominent presence of red regions along the diagonal, which corresponds to its superior average accuracy compared to FCS.

## E DISCUSSION

**Comparison with prompt-based method**. Unlike prompt-based approaches that operate by tuning only prompt parameters while keeping the pre-trained models fixed (Lu et al., 2025; Nguyen et al., 2025), our method takes a fundamentally different path by updating the backbone throughout con-

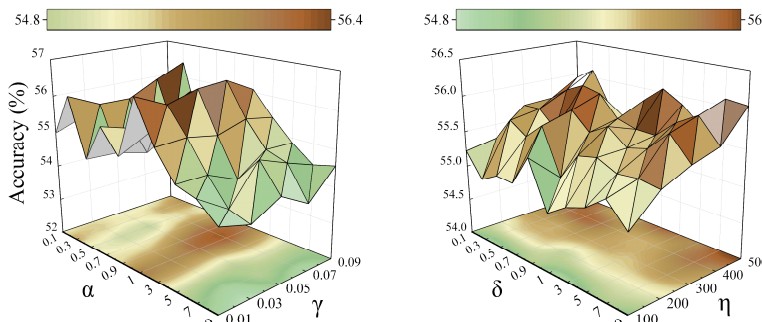

Figure 10: The impact of the balancing parameters on the last task accuracy in the 5-phase partition of CIFAR-100. The left figure shows the relationship between $\alpha$ and $\gamma$, while the right figure presents the correlation between $\delta$ and $\eta$.

tinual learning. For instance, HiDe-Prompt (Wang et al., 2023) maintains frozen Vision Transformer backbones and employs a prompt ensemble mechanism, resulting in static feature representations. In contrast, our approach incrementally refines the backbone to acquire new representations for each task. Since these prompt-based strategies avoid feature drift by design, they are not directly addressing the core challenge targeted in our work. Moreover, directly comparing such frozen models with our from-scratch training paradigm would be inappropriate. Although fixed backbones perform well on widely used datasets, real-world scenarios with domain-specific or out-of-distribution data demand continuous backbone adaptation, highlighting the necessity of effective drift compensation techniques.

**Integration with Pre-trained Model**. Pre-trained models have been increasingly adopted for continual incremental learning (CIL). Our method can be readily combined with such models, including widely used architectures like ViT (Dosovitskiy et al., 2021). To facilitate the adaptation to novel classes while utilizing the frozen knowledge in pre-trained models, a lightweight LoRA (Low-Rank Adaptation) (Hu et al., 2022) module can be incorporated into each transformer block. This module introduces two low-rank matrices $\mathbf{G} \in \mathbb{R}^{r \times v}$, $\mathbf{H} \in \mathbb{R}^{u \times r}$, where $r \ll \min(u, v)$ indicates the rank. Let $\tilde{z} \in \mathbb{R}^v$ represent the intermediate features of the current layer. The forward computation with LoRA is modified as:

$$z' = (\mathbf{A} + \mathbf{HG})\,\tilde{z} = \mathbf{A}\tilde{z} + \mathbf{HG}\tilde{z},$$

where $z' \in \mathbb{R}^u$ denotes the output passed to the next layer following a non-linear activation. During training, the original weight matrix $\mathbf{A} \in \mathbb{R}^{u \times v}$ remains fixed and only the low-rank matrices $\mathbf{H}$ and $\mathbf{G}$ are updated. This makes the method both parameter-efficient and computationally economical. In the CIL framework, LoRA parameters evolve with each task, which may lead to feature drift. Therefore, the drift compensation method proposed in this paper is also applicable to pre-trained models.

## THE USE OF LARGE LANGUAGE MODELS (LLMS)

In this work, LLMs were used solely as a general-purpose tool to improve the clarity, fluency, and readability of the English text. LLMs were not involved in the research ideation, experimental design, data analysis, or interpretation of results.

