# OpenReview forum: "Evoking Generalized Cognition for Exemplar Free Continual Learning via Granular Ball Representation"
_ICLR.cc/2026/Conference — ICLR 2026 Conference Withdrawn Submission_

### Official Review · Reviewer_8nUi · 2025-10-17

**Soundness:** 2
**Presentation:** 3
**Contribution:** 3
**Rating:** 4
**Confidence:** 5

**Summary:**

Authors implement exemplar-free class-incremental learning method that represents each class with multi-granularity “granular balls” (centers + radii + labels) and progressively “blurs” old classes to counter growing granularity of newer ones, plus a lightweight Synergy Drift Estimation (SDE) to keep past concepts aligned as the feature space shifts. The methods shows improvements in mitigating the catastrophic forgetting when on various benchmarks.

**Strengths:**

1) Multi-Granularity Representation Improves Knowledge Retention
BallIL introduces granular balls to represent each class at multiple levels of abstraction (center + radius).
This design captures both fine and coarse class features, enabling the model to generalize better and preserve older class knowledge without relying on stored exemplars. It directly addresses the imbalance of class granularity that usually leads to forgetting in continual learning.

2) Synergy Drift Estimation (SDE) Reduces Representation Drift
The SDE module dynamically updates older class representations as the feature extractor evolves. This continuous alignment ensures that past knowledge remains compatible with the new embedding space, reducing catastrophic forgetting and improving cross-task stability.

3) Introduces new concept to continual to mitigate the forgetting. The entire text the well articulated. Easy to follow.

**Weaknesses:**

1) When the blur factor (the mechanism used to expand old class representation) is set too high, model performance declines, both in terms of last task accuracy and average incremental accuracy. This suggests that while expanding the representation of old classes is helpful up to a point, too much expansion over-corrects for class imbalances and actually harms the system’s ability to retain finer distinctions.

3) High Parameter Sensitivity and Training Complexity
BallIL relies on several hyperparameters — such as the blur factor (ψ), uncertainty weights (α, γ, η, δ) — that must be carefully tuned. Incorrect settings can easily lead to over-generalization or underfitting, making the method less practical for large-scale or fast-deploy scenarios.

3) Simplistic Assumption of Spherical Class Boundaries
Each class is represented as a sphere (ball) in feature space, assuming uniform spread in all directions. This may fail to model complex or anisotropic class distributions, especially in real-world, high-dimensional data where class shapes are irregular or multimodal

**Questions:**

I have few questions:

1) Synergy Drift Estimation seems like a feature distillation between current and previous task scaled with A^{SDE} ?

2) Though you mentioned about the datasets, its unclear that how the datasets are split in Table 1. Did you use 50% of the classes in first task and divide the rest into 5 (20 classes / task )and 10 phases? or whole classes are divided into 5 and 10 phases? Seeing the FeCAM results, I assume it is cold start. It is recommend to compare how the method would stand when comparing with large initial classes.

3) In implementation details, you mentioned different hyper-params for initial and subsequent task for example epochs and learning rate. What is the difference?
For example:
You train 0-9 with 200 epochs and 10-19, 20-29.....with 100 epochs? same goes with other.
This would make sense if the Table 1 is Warm start --- training 0-49 for the 200 epochs and 50-59 and so on.

3) Though it is better than some methods, there are other powerful methods such as EFC[1], LDC [2], ADC[3], AdaGauss[4] and authors should compare with them. These methods really excel in EFCIL in cold start learning.

[1] Magistri, Simone, et al. "Elastic feature consolidation for cold start exemplar-free incremental learning." arXiv preprint arXiv:2402.03917 (2024).

[2] Gomez-Villa, Alex, et al. "Exemplar-free continual representation learning via learnable drift compensation." European Conference on Computer Vision. Cham: Springer Nature Switzerland, 2024.

[3] Goswami, Dipam, et al. "Resurrecting old classes with new data for exemplar-free continual learning." Proceedings of the IEEE/CVF Conference on Computer Vision and Pattern Recognition. 2024.

[4] Rypeść, Grzegorz, et al. "Task-recency bias strikes back: Adapting covariances in exemplar-free class incremental learning." Advances in Neural Information Processing Systems 37 (2024): 63268-63289.

---

### Official Review · Reviewer_tzLL · 2025-10-31

**Soundness:** 3
**Presentation:** 3
**Contribution:** 2
**Rating:** 6
**Confidence:** 5

**Summary:**

This paper introduces BallIL, a new method for Exemplar-Free Class-Incremental Learning (EFCIL) that tackles catastrophic forgetting without storing old samples. The key idea is to represent classes using granular balls which are adaptive, multi-granularity abstractions that generalize past concepts over time. The method gradually broadens older class representations using a blur factor to keep them consistent with newer tasks. It adds losses that measure how uncertain the model is in recognizing old versus new classes, helping reduce confusion. Finally, a Synergy Drift Estimation (SDE) module keeps old class knowledge up to date as the model’s features change over time.

**Strengths:**

1. The use of multi-granularity representation via granular balls is conceptually new for continual learning, bridging granular computing and EFCIL.
2. Ablations clarify contributions of each component across different datasets.

**Weaknesses:**

1. While the paper offers a novel application of granular ball theory, the claim of being the first to approach EFCIL from a multi-granularity perspective should be toned down. Hierarchical continual learning methods have previously addressed learning and forgetting from different levels of granularity.
2. While empirical results are strong, theoretical guarantees (e.g., on granularity adaptation or uncertainty estimation) are not provided.
3. Considered a very simple setting of continual learning and omitted i-Blurry[1], SI-Blurry[2] and real world continual learning[3] settings which are more promising.
4. Omitted the performance comparison against the strongest methods such as ADC[4], LDC[5] which significantly weakens the empirical claim of the work.

Citations:
[1]Online Continual Learning on Class Incremental Blurry Task Configuration with Anytime Inference (Hyunseo Koh, Dahyun Kim, Jung-Woo Ha, Jonghyun Choi)
[2]Online Class Incremental Learning on Stochastic Blurry Task Boundary via Mask and Visual Prompt Tuning (Jun-Yeong Moon, Keon-Hee Park, Jung Uk Kim, Gyeong-Moon Park)
[3] Online Class-Incremental Learning For Real-World Food Image Classification (Siddeshwar Raghavan, Jiangpeng He, Fengqing Zhu)
[4] Resurrecting Old Classes with New Data for Exemplar-Free Continual Learning (Dipam Goswami, Albin Soutif--Cormerais, Yuyang Liu, Sandesh Kamath, Bartłomiej Twardowski, Joost van de Weijer)
[5]Exemplar-free Continual Representation Learning via Learnable Drift Compensation (Alex Gomez-Villa, Dipam Goswami, Kai Wang, Andrew D. Bagdanov, Bartlomiej Twardowski, Joost van de Weijer)

**Questions:**

1. What is the visual or conceptual meaning of “granularity” in deep feature space? It could use clearer qualitative analysis.
2. Could the authors clarify the relationship between granularity generalization and existing prototype-based drift compensation methods?
3. How does BallIL behave when tasks are semantically dissimilar?
4. What is the intuition behind linking representation/decision uncertainty to Shannon entropy. Does this have empirical or theoretical grounding?

I am willing to change my review if the concerns are addressed.

---

### Official Review · Reviewer_p2Yb · 2025-10-31

**Soundness:** 3
**Presentation:** 3
**Contribution:** 3
**Rating:** 4
**Confidence:** 4

**Summary:**

This paper addresses catastrophic forgetting in Exemplar-Free Class-Incremental Learning (EFCIL) by proposing Granular Ball Incremental Learning (BallIL), which leverages granular ball representation to capture multi-granularity class characteristics and progressively expands the granularity of old classes to mitigate bias toward new classes. BallIL further introduces a Synergy Drift Estimation (SDE) module to dynamically update outdated old-class concepts in new feature spaces, along with concept-informed uncertainty losses, and achieves state-of-the-art performance across six benchmark datasets (e.g., CIFAR-100, ImageNet-1K).

**Strengths:**

1. The work is the first to tackle EFCIL from a multi-granularity perspective, identifying that increasing class granularity with task progression exacerbates forgetting—an insightful observation that fills a gap in existing single-fixed-granularity EFCIL methods.

2. The integration of SDE (for real-time drift compensation) and concept-informed representation/decision uncertainty losses is well-designed: it not only preserves old-class concepts amid feature drift but also quantifies and alleviates confusion between old and new classes, with rigorous ablation studies validating each component’s effectiveness.

**Weaknesses:**

1. The paper only briefly mentions combining BallIL with pre-trained models (e.g., ViT via LoRA) but provides no experimental results on this integration. It remains unclear how BallIL’s granularity mechanisms interact with pre-trained features, nor whether it maintains advantages in parameter-efficient incremental learning setups.

2. It is suggested to supplement experiments that measure the computational complexity and memory costs of BallIL and compare it with the SOTA methods to verify its efficiency and practicality.

3. It is suggested to add the forgetting rate metric and compare it with the SOTA methods to directly evaluate BallIL’s effectiveness in alleviating catastrophic forgetting.

4. The paper lacks formal analysis of generalization bounds or convergence guarantees for the granular ball representation, weakening the theoretical foundation.

**Questions:**

See weakness.

---

### Official Review · Reviewer_Br8X · 2025-11-03

**Soundness:** 2
**Presentation:** 3
**Contribution:** 2
**Rating:** 4
**Confidence:** 4

**Summary:**

The paper studies exemplar-free class-incremental learning and uses granular ball representations to develop a concept-informed representation and decision uncertainty evaluations for classification tasks. The method also uses a drift estimation module to update past class concepts in new tasks. The proposed method performs well on several settings across datasets.

**Strengths:**

1. The paper is well-written with an interesting approach of using granular ball representations for EFCIL.
2. The method is intuitive and novel.

**Weaknesses:**

1. The paper lacks discussions and strong motivations for using granular ball representations over approaches using feature covariances (AdaGauss, FeCAM).

2. The method uses several hyper-parameters with no proper explanation of how these values are selected.

3. The experiments ignores several recent competitive approaches for EFCIL like ADC [a], LDC [b], EFC [c], AdaGauss [d] (Table 1, Fig. 4). These methods already outperforms the proposed method in several settings.

[a] “Resurrecting old classes with new data for exemplar-free continual learning." Proceedings of the IEEE/CVF Conference on Computer Vision and Pattern Recognition. 2024.

[b] “Exemplar-free continual representation learning via learnable drift compensation." European Conference on Computer Vision. Cham: Springer Nature Switzerland, 2024.

[c] "Elastic Feature Consolidation For Cold Start Exemplar-Free Incremental Learning." The Twelfth International Conference on Learning Representations.

[d] "Task-recency bias strikes back: Adapting covariances in exemplar-free class incremental learning." Advances in Neural Information Processing Systems, 2024.

**Questions:**

1. How is the synergy drift estimation module different from LDC? LDC proposed to learn and map prototypes to new space after the incremental learning phase. Here, the authors do the same. Can you provide more clarity on the proposed “joint cross-training strategy”?

2. Can the authors discuss how considering the granular ball representation is beneficial instead of using the class-wise feature covariances for granularity? For instance, Fecam [x] shows similar analysis that new classes have more scattered anisotropic covariances compared to old classes.

[x] Fecam: Exploiting the heterogeneity of class distributions in exemplar-free continual learning. In Advances in Neural Information Processing Systems, 2023.

---

### Note · Authors · 2026-02-28

I have read and agree with the venue's withdrawal policy on behalf of myself and my co-authors.

---

### Meta-Review · Area_Chair_mfho · 2025-12-09

**Summary:**

The paper addresses catastrophic forgetting in Exemplar-Free Class-Incremental Learning. It introduces Granular Ball Incremental Learning (BallIL), which uses granular ball representations for class descriptions and expands the granularity of old classes to mitigate the effects of feature drift and balance inter-task differences. The authors claim that BallIL enhances classification performance, supported by experiments conducted across multiple datasets.

While reviewers appreciated the clarity, novelty, and provided ablations, they note a lack of robust discussion on the advantages of granular ball representations compared to existing methods and a failure to provide sufficient motivation for BallIL. They also note that several competitive models from recent literature are not included in the experimental comparisons, which weakens the claim of superior performance. Additionally, they point to insufficient analysis of the hyperparameter selection process, which raises concerns about the practicality in real-world scenarios. Issues related to high sensitivity to hyperparameters and simplistic assumptions about class representations were also highlighted by reviewers, suggesting practical limitations. Overall, these shortcomings undermine the potential impact of the work. The authors did not provide a rebuttal, and the recommendation is to Reject.

**Reviewer Concerns:**

The authors provided no rebuttal.

**Reviewer Scores:**

The authors provided no rebuttal, but given the weaknesses unanimously pointed out by the reviewers I do not believe they would have been convinced to improve their scores.

---

### Decision · Program_Chairs · 2026-01-26

Reject